# Rapid Removal of Organic Pollutants from Aqueous Systems under Solar Irradiation Using ZrO_2_/Fe_3_O_4_ Nanoparticles

**DOI:** 10.3390/molecules27228060

**Published:** 2022-11-20

**Authors:** Nemanja Banić, Daniela Šojić Merkulov, Vesna Despotović, Nina Finčur, Tamara Ivetić, Szabolcs Bognár, Dušica Jovanović, Biljana Abramović

**Affiliations:** 1Department of Chemistry, Biochemistry and Environmental Protection, University of Novi Sad Faculty of Sciences, Trg Dositeja Obradovića 3, 21000 Novi Sad, Serbia; 2Department of Physics, University of Novi Sad Faculty of Sciences, Trg Dositeja Obradovića 4, 21000 Novi Sad, Serbia

**Keywords:** heterogeneous photocatalysis, organic pollutants, solar irradiation, ZrO_2_/Fe_3_O_4_ nanoparticles, aqueous environment

## Abstract

Pure water scarcity is an emerging, all-around problem that globally affects both the life quality and the world’s economy. Heterogeneous photocatalysis under solar irradiation is a promising technique for the organic pollutants (e.g., pesticides, drugs) removal from an aqueous environment. Furthermore, the drawbacks of commercially available photocatalysts can be successfully overcome by using innovative nanoparticles, such as ZrO_2_/Fe_3_O_4_. Four ZrO_2_/Fe_3_O_4_ nanopowders with a different mass ratio of ZrO_2_ and Fe_3_O_4_ were synthesized using the chemical co-precipitation method. XRD analysis showed the presence of magnetite and hematite Fe-oxide phases in all samples. The content of the magnetite phase increased with the addition of 19% ZrO_2_. The efficiency of the newly synthesized ZrO_2_/Fe_3_O_4_ nanoparticles was investigated in the rapid removal of selected pollutants under various experimental conditions. Nevertheless, the influence of the water matrix on photocatalytic degradation was also examined. The obtained data showed that using ZrO_2_/Fe_3_O_4_ nanosystems, an appropriate removal rate of the selected pesticides and pharmaceuticals can be reached after 120 min of solar irradiation. Further, the total organic carbon measurements proved the mineralization of the target emerging pollutants. ZrO_2_/Fe_3_O_4_ nanoparticles are economically feasible, as their removal from the suspension can be easily achieved using affordable, environmentally-friendly magnetic separation.

## 1. Introduction

Nowadays, environmental pollution with persistent, emerging contaminants, such as pesticides and active pharmaceutical ingredients (APIs), is a great global concern. Furthermore, organic dyes are also harmful compounds to the environment due to the possible diseases they can cause in human beings, like skin dermatitis, allergies, etc. [1]. The lack of adequate and effective methods for their complete removal also adds up to this serious, comprehensive issue [2]. All natural resources are affected by environmental pollution. However, water contamination is the most significant. It is highlighted that in middle-income countries, GDP growth falls by half in areas downstream of highly-polluted water bodies [3]. In addition, the worldwide production of these micro-pollutants has reached around 500 million tons/year [4]. Different pesticides and APIs were detected in natural water ecosystems in the concentration range of μg/dm^3^ [5]. The intensive application of pesticides in agriculture leads to serious contamination of soil and water resources [6].

Unfortunately, conventional wastewater treatment techniques, which are mainly based on different chemical and biological approaches, are not capable of removing the mentioned organic pollutants [7]. Moreover, each treatment technique has its own advantages and limitations not only in terms of capital and operational costs but also in terms of efficiency, operability, reliability, environmental impact, pre-treatment requirements, and the production of sludge and toxic byproducts [4]. Hence, there is a need to find alternative, green and competent removal methods.

Advanced oxidation processes (AOPs) are promising alternatives to the earlier-mentioned conventional techniques. The main power of the AOPs lies in the formation of reactive oxidative species followed by hydrogen distraction, the introduction of hydroxyl groups or electron shift. Besides the above-mentioned, other simultaneously formed reactive oxidative species, for instance, superoxide ions, peroxymonosulfate ions, hydrogen ions, sulfate ions, hydroperoxyl ions, and singlet ions, also participate in the AOPs-based degradation techniques [8]. Heterogeneous photocatalysis is one of the AOPs. For a successful photocatalytic process, it is necessary to have an adequate source of irradiation and a semiconductor as a photocatalyst. Every photocatalyst contains valance and conduction bands, which are separated by a bandgap. In order to come to a photocatalytic reaction, the energy of photons absorbed by the semiconductors has to be equal to or higher than its bandgap energy. There are different semiconductors applied as photocatalysts, such as TiO_2_, SnO_2_, SiO_2_, CeO_2_, ZnO, WO_3_ and ZrO_2_ [9]. Unfortunately, many drawbacks are also observed, such as the recombination of the photogenerated electron-hole pairs, low heat sensitivity and selectivity [10]. Nevertheless, recently various biodegradation processes, such as biodegradation by *Pseudomonas aeruginosa*, are also investigated as potential remediation techniques [11,12,13].

Currently, great attention is paid to the improvement of photocatalytic activity by different nanopowders, which are able to modify the surface [14]. ZrO_2_ nanoparticles are interesting due to their increased optical and electrical properties [15]. The ZrO_2_ is used as a photocatalyst in heterogeneous reactions due to its semiconductor properties (type *n*). The bandgap energy of ZrO_2_ varies from 3.25 to 5.1 eV depending on the sample synthesis. Its common bandgap energy is about 5 eV. Mansouri et al. examined the use of ZrO_2_ zeolite nanoparticles for methyl orange removal in the presence of UV radiation and observed more than 97% of dye removed during 80 min [16]. Basahel et al. reported the application of monoclinic ZrO_2_ nanoparticles in methyl orange removal under the influence of UV irradiation; more than 95% dye was removed within 110 min [17]. Saeed et al. studied the ZrO_2_-supported palladium and platinum nanoparticles in the photodegradation of indigodisulfonate; they established the percentage degradation as 96 and 94% within 14 h, respectively [18]. The magnetic behavior of Fe_3_O_4_ nanoparticles makes recyclable nanocomposites possible. Furthermore, the excitation range of nanocomposites synthesized from Fe_3_O_4_ can be used in the visible light range due to their unique optical properties [19].

The aim of this study is to investigate the efficiency of photodegradation of pesticides (sulcotrione and fluroxypyr herbicide, as well as thiacloprid insecticide) and API (amitriptyline) in the presence of unmodified Fe_3_O_4_, as well as ZrO_2_-modified Fe_3_O_4_ nanopowders synthesized by co-precipitation method with a different mass ratio of ZrO_2_:Fe_3_O_4_, using solar irradiation. X-ray diffraction (XRD), Raman, scanning electron microscopy with energy dispersive spectroscopy (SEM–EDS) and diffuse reflectance spectroscopy (DRS) are used for the characterization of the synthesized photocatalysts. Furthermore, the effect of water matrix (ultrapure, river (Danube), underground and drinking water) on the efficiency of photocatalytic degradation was also examined.

## 2. Results and Discussion

### 2.1. SEM and EDS Analysis of ZrO_2_/Fe_3_O_4_ Catalysts

Figure 1 and Appendix A show SEM images of unmodified Fe_3_O_4_ (hereafter “Fe_3_O_4_”) and unsupported ZrO_2_ (hereafter “ZrO_2_”) particles (which were annealed at 300 °C to be prepared at the same temperature as the synthesized ZrO_2_/Fe_3_O_4_ nanopowders), as well as all four synthesized ZrO_2_/Fe_3_O_4_ nanopowders. The difference in morphology of Fe_3_O_4_ (Figure 1a or Appendix A) and ZrO_2_ samples (Figure 1b or Appendix A) is quite obvious. In contrast, the difference in particle size between Fe_3_O_4_ and synthesized ZrO_2_/Fe_3_O_4_ nanopowders with higher ZrO_2_ content (3.5, 12, and 19%) is small (Figure 1d–f). On the other hand, in the case of 0.9ZrO_2_/Fe_3_O_4_ (Figure 1c), when the particle sizes vary in a broader range, SEM images of the synthesized ZrO_2_/Fe_3_O_4_ nanopowders show that the particles build clusters of irregular shapes whose size varies between 55 and 198 nm.

The chemical composition of synthesized nanopowders has been estimated by the EDS method. The EDS spectra are collected from corresponding framed areas of SEM images, shown together in Figure 2. As may be noted, zirconium was identified in all samples (Figure 2c,d). However, in the case of the lowest content, which is given in Table 1, zirconium was not quantified.

### 2.2. XRD and Raman Measurements of ZrO_2_/Fe_3_O_4_ Catalysts

XRD patterns of Fe_3_O_4_, synthesized ZrO_2_/Fe_3_O_4_ catalysts, and ZrO_2_ are shown in Appendix A. The XRD peak intensities of ZrO_2_ were reduced eight times from their original values to be comparable to the XRD patterns of the rest of the samples and indexed as the baddeleyite ZrO_2_ phase according to ICDD card no. 01-072-1669, with an average crystallite size of 61 nm. The presence of the ZrO_2_ phase was confirmed in all ZrO_2_/Fe_3_O_4_ catalysts, and as expected, the intensity and number of XRD peaks increased with increasing ZrO_2_ concentration during synthesis (Appendix A). Two Fe-oxide phases were identified in the XRD patterns of the mixed catalysts, magnetite, Fe_3_O_4_, and the hematite, Fe_2_O_3_, according to ICDD card no. 01-075-0449 and 01-073-2234, respectively. The calculated crystallite sizes of the two Fe-oxide phases, as well as their phase ratios, are given in Table 2. Synthesized by the same procedure, the Fe-oxide material (without ZrO_2_) also has two Fe-oxide present phases, hematite (α-Fe_2_O_3_) and magnetite (Fe_3_O_4_), in the ratio of about 0.5 to 1. The addition of up to 12% ZrO_2_ affects the ratio of Fe-oxide phases (α-Fe_2_O_3_/Fe_3_O_4_) by increasing the hematite phase, while the largest addition of 19% ZrO_2_ causes an increase in the magnetite phase compared to the ratio of Fe-oxide phases in the Fe-oxide material. The size of Fe_3_O_4_ crystallites does not change significantly and is approximately 14 nm. The largest Fe_3_O_4_ crystallite size of 16 nm is achieved with the addition of 19% ZrO_2_. The size of the hematite crystallites is approximately 6 nm and reaches a maximum value of about 8 nm with the addition of 19% ZrO_2_.

Raman spectroscopy (Appendix A) is used for further analysis of the iron oxide phase transition. Group theory predicts five Raman active vibration modes of the magnetite phase (*A*_1g_ + *E*_g_ + 3*T*_2g_) and seven structural modes of the hematite phase (2*A*_1g_ + 5*E*_g_) [20]. The Raman spectra of the ZrO_2_/Fe_3_O_4_ catalysts are similar to the Raman spectrum of the Fe_3_O_4_ sample showing 5 peaks in the range of 100–800 cm^−1^. The most intense peaks are found at around 214 cm^−1^ and 275 cm^−1^, followed by three peaks of lower intensity at around 383 cm^−1^, 481 cm^−1^, and 582 cm^−1^. Similar reports are found in the literature for the Raman spectrum of Fe_3_O_4_ nanoparticles [21,22] shaped by the transformation of magnetite into hematite due to the high laser power used in Raman experiments [21,23]. No Raman peaks of the ZrO_2_ phase structure were found in the ZrO_2_/Fe_3_O_4_ samples. The intensity of the ZrO_2_ Raman spectrum is reduced by half and is shown in Appendix A for comparison.

### 2.3. UV-Vis DRS of ZrO_2_/Fe_3_O_4_ Catalysts

The DRS spectra (Figure 3) show that the absorption of the synthesized ZrO_2_/Fe_3_O_4_ catalysts is similar to Fe_3_O_4_ and significantly higher than ZrO_2_ in the observed wavelength range indicating their high photocatalytic activity under visible irradiation. Furthermore, DRS was used to estimate the optical bandgap energy (*E*_g_) by plotting the (*F*(*R*)·*hν*)^1/*2*^ against the photon energy (*hν*), where *F*(*R*) is the Kubelka-Munk transformation of the measured diffuse reflectance (%) defined as (1-*R*)^2^/2*R* (Appendix A) [24]. Estimated values are 2.22 eV for Fe_3_O_4_, 2.23 eV for 0.9ZrO_2_/Fe_3_O_4_ and 3.5ZrO_2_/Fe_3_O_4_, and 2.24 eV for 12ZrO_2_/Fe_3_O_4_ and 19ZrO_2_/Fe_3_O_4_ samples, and are typically expected values of energy gap for iron oxide nanoparticles, 2–3 eV [25]. The optical absorption threshold, then calculated by λ = 1240/*E*_g_, is around 558 nm, 556 nm, and 553 nm, respectively, confirming the vast visible absorption capacity of these samples for photocatalysis.

### 2.4. The Efficiency of Photocatalytic Degradation of the Selected Pesticides and API Using ZrO_2_/Fe_3_O_4_ Nanopowders

It is believed that photochemical decomposition is one of the most efficient processes used to remove organic pollutants from the environment. Many of these compounds in their structure contain aromatic cores, heteroatoms and other functional groups that can react with substances present in natural waters [26]. Furthermore, the presence of dissolved organic matter and various inorganic ions in natural waters also affects the efficiency of the photocatalytic degradation of organic pollutants [27,28,29].

#### 2.4.1. Photodegradation of Pesticides

In this research, the efficiency of photocatalytic degradation of three pesticides (thiacloprid, sulcotrione and fluroxypyr) and one API (amitriptyline) was investigated.

Thiacloprid. Thiacloprid belongs to the new group of insecticides. Studies about the behavior of this pollutant in the environment showed that this compound is persistent to hydrolysis in the acidic and neutral environment for more than six months, while at pH = 10, only 10% of thiacloprid can be degraded in aerated water, after a six months period [30,31]. The photodegradation efficiency of thiacloprid was studied under different experimental conditions, using various AOPs under solar irradiation (Figure 4). Based on the obtained results (Figure 4), it can be seen that the increased content of ZrO_2_ in the newly synthesized nanopowders leads to enhanced degradation efficiency. Moreover, experiments were also carried out in the presence of H_2_O_2_. The presence of H_2_O_2_ can enhance the efficiency of the photocatalytic process in two ways; as an electron scavenger or as a component of the photo-Fenton process [32]. The highest degradation efficiency of thiacloprid was reached in the system with 12ZrO_2_/Fe_3_O_4_ and H_2_O_2_. Namely, 35% of thiacloprid degraded after 60 min of solar irradiation. Since 12ZrO_2_/Fe_3_O_4_ and 19ZrO_2_/Fe_3_O_4_ nanopowders showed almost the same efficiency, 12ZrO_2_/Fe_3_O_4_ was used in further experiments because of the lower content of ZrO_2_ (Figure 4).

By comparing the kinetic curves in Figure 4, it may be seen that using a heterogeneous 12ZrO_2_/Fe_3_O_4_/H_2_O_2_ photo-Fenton system, the efficiency of thiacloprid photodegradation was about 1.3 times higher compared to Fe_3_O_4_/H_2_O_2_. From these results, it can be concluded that besides the influence of the heterogeneous photo-Fenton process on the degradation efficiency, the coupling effect of the semiconductor also has a significant influence [33]. This makes ZrO_2_/Fe_3_O_4_ materials more suitable for application than unmodified Fe_3_O_4_. 

Furthermore, the influence of the initial H_2_O_2_ concentration on the thiacloprid degradation efficiency was also examined in the concentration range from 6.43 to 67.5 mmol/dm^3^ (Figure 5a). Based on the obtained data, it can be seen that the efficiency was increasing until the H_2_O_2_ concentration reached 45 mmol/dm^3^. On the other hand, a further increase in H_2_O_2_ concentration did not have an effect on the degradation. This different effect of the higher H_2_O_2_ concentration is caused by two competing processes. Firstly, the increased H_2_O_2_ concentration leads to a higher number of ^•^OH-radicals which can successfully attack thiacloprid. Secondly, H_2_O_2_ can also act as a scavenger of ^•^OH-radicals, whereby less reactive hydroperoxyl-radicals are formed (reactions 1 and 2). This influence becomes more intensive at higher concentrations of H_2_O_2_ since less ^•^OH-radicals are available for the degradation of selected pesticides [34].
(1)$$H_{2}O_{2}+{\;}^{•}\mathrm{OH}\;\to \;H_{2}O+{\mathrm{HO}}_{2}^{•}$$
(2)$${\mathrm{HO}}_{2}^{•}+{\;}^{•}\mathrm{OH}\;\to \;H_{2}O+O_{2}$$

In addition, the influence of the initial pH value of the suspension (2.8–7.8) was also investigated (Figure 5b). Based on the findings, it can be seen that pH 2.8 was optimal, similar to the photocatalytic degradation of thiacloprid in the presence of Fe/TiO_2_ [35]. This system is proved to be efficient for generating radicals for oxidative processes, especially at pH 2.8, where approximately one-half of the Fe(III) is present as the Fe^3+^ ion and the other half as Fe(OH)^2+^ ion: as photoactive species [36].

Moreover, the influence of the 12ZrO_2_/Fe_3_O_4_ photocatalyst loading (from 0.21 to 1.67 mg/cm^3^) on the efficiency of thiacloprid photodegradation was also examined (Figure 6). Increased degradation efficiency was observed up to 0.83 mg/cm^3^ catalyst loading, while the efficiency was reduced at higher loading. The enhanced efficiency can be explained by the fact that at higher photocatalyst loading, there is an increased number of available active sites on the catalyst’s surface. On another note, further increase leads to decreased efficiency due to more intense opacity of the suspension and radiation scattering, which reduces the amount of radiation through the suspension [37].

It is well known that the separation of the catalyst nanoparticles from reaction suspensions requires significant financial investments. On the other hand, the application of magnetic nanoparticles offers simple and fast separation since they can be easily collected by applying an external magnetic field [38]. Based on the mentioned, the magnetic separation of the newly synthesized 12ZrO_2_/Fe_3_O_4_ nanopowder from the suspension was also investigated (Figure 7). Our findings indicate that the photocatalysts can be successfully separated under the influence of a permanent magnet.

Sulcotrione. Another pesticide that was investigated in this study is sulcotrione. Sulcotrione belongs to the newer group of triketone herbicides, and it is used for wide spectra grass and broad-leaved weed control in maize fields [39]. This herbicide is soluble in water, so it can easily reach the surface and underground water resources. This pollutant is persistent in an aqueous environment both in the presence and absence of sunlight [40]. The photodegradation of sulcotrione was investigated in the presence of Fe_3_O_4_ and ZrO_2_/Fe_3_O_4_ (3.5, 12, and 19%) nanopowders with 45 mmol/dm^3^ (Figure 8a) and 3.0 mmol/dm^3^ (Appendix A) H_2_O_2_, at pH 2.8. Firstly, in the presence of the above-mentioned catalysts, a great adsorption rate was observed during the sonication process in the dark, with both molar concentrations of H_2_O_2_. The highest adsorption (~98%) was found in the presence of Fe_3_O_4_ with 45 mmol/dm^3^ H_2_O_2_ after 15 min of sonication in the dark (Figure 8a). However, in the case of 12ZrO_2_/Fe_3_O_4_ and 19ZrO_2_/Fe_3_O_4_ nanopowders, lower adsorption was detected, namely 61% and 73%, respectively. Furthermore, in the systems with Fe_3_O_4_ and 19ZrO_2_/Fe_3_O_4_, desorption of sulcotrione was detected during the process of photodegradation. The percentage of adsorption and degradation in the presence of Fe_3_O_4_, 12ZrO_2_/Fe_3_O_4_ and 19ZrO_2_/Fe_3_O_4_ with 45 mmol/dm^3^ H_2_O_2_ was 97%, 94%, and 85% after 240 min of irradiation, respectively. Similarly, in the systems with 3.0 mmol/dm^3^ of H_2_O_2_ adsorption was also monitored on the investigated nanopowders with the following percentage: 82%, 53%, and 78%, in the case of 3.5, 12, and 19ZrO_2_/Fe_3_O_4_ (Appendix A). In addition, in the presence of all three coupled ZrO_2_/Fe_3_O_4_ catalysts, both adsorption and desorption were detected during the photodegradation process. Namely, in the presence of 3.0 mmol/dm^3^ H_2_O_2_ there was 92%, 52%, and 98% of sulcotrione removed with 3.5ZrO_2_/Fe_3_O_4_, 12ZrO_2_/Fe_3_O_4_, and 19ZrO_2_/Fe_3_O_4_ respectively, after 240 min of irradiation.

Since the sonication had a strong influence on the reaction system, the effect of stirring prior to irradiation was also investigated. Experiments were carried out in the systems with Fe_3_O_4_ and ZrO_2_/Fe_3_O_4_ (3.5, 12, and 19%) with 3.0 mmol/dm^3^ H_2_O_2_ at pH 2.8, using solar irradiation (Appendix A). The stirring process was performed 15 min prior to the irradiation, and the obtained results showed that the adsorption rate was lower compared to the sonication process. Namely, the lowest adsorption was monitored in the system with Fe_3_O_4_ (~30%), while in the presence of 3.5ZrO_2_/Fe_3_O_4_, 12ZrO_2_/Fe_3_O_4_, and 19ZrO_2_/Fe_3_O_4_ nanopowders the adsorption was slightly higher, more preciously 35%, 34%, and 38%, respectively. Furthermore, the total removal efficiency of sulcotrione in the presence of Fe_3_O_4_, 3.5ZrO_2_/Fe_3_O_4_, 12ZrO_2_/Fe_3_O_4_, and 19ZrO_2_/Fe_3_O_4_ at pH 2.8 and with 3.0 mmol/dm^3^ H_2_O_2_ was also examined, including both the adsorption and photodegradation process. Based on the obtained data, it can be concluded that the highest removal rate was achieved in the system with Fe_3_O_4_ (99%), while in the case of 3.5ZrO_2_/Fe_3_O_4_, 12ZrO_2_/Fe_3_O_4_, and 19ZrO_2_/Fe_3_O_4_ there were 93%, 88%, and 96% of sulcotrione removed, respectively. Moreover, our findings also showed that under the above-mentioned experimental conditions, the ZrO_2_ coupled catalysts did not have higher efficiency in the degradation of the mentioned pesticide compared to the pure Fe_3_O_4_.

Based on the above, it can be concluded that substrate structure has a significant influence on the efficiency of photodegradation, as well as the adsorption of the substrate on nanopowders. Namely, in the case of thiacloprid, there is no adsorption, and only photocatalytic degradation takes place, while in the case of sulcotrione, adsorption is very pronounced in comparison to photocatalysis.

Last but not least, total organic carbon (TOC) measurements were also conducted in order to determine the efficiency of the removal process. Based on the TOC findings, it can be concluded that the highest mineralization rate was achieved in the system with 12ZrO_2_/Fe_3_O_4_ when 17.4% of sulcotrione was mineralized after 240 min of solar irradiation. These results indicate that the photodegradation of sulcotrione takes place via various degradation intermediates.

Fluroxypyr. The photocatalytic degradation of fluroxypyr was also examined in this research. Fluroxypyr is a derivate of pyridinic acid, and it is a translocated herbicide that is used to control broadleaf weeds and woody brush in small grain cereals, maize, pastures, rangeland, and in fruit and wine grapes plantations. The action of fluroxypyr is based on the auxin-type responses as an indol acetic acid and leads to the growth of tissues to higher concentrations than the native auxin does and also degrades more slowly. As a result, it comes to characteristic deformation and curling of plant tissues [41]. In order to study the elimination efficiency of herbicide fluroxypyr from water, it was found that increasing the amount of ZrO_2_, in the range from 3.5 to 19% (*w/w*) in newly synthesized nanopowders based on Fe_3_O_4_, leads to increased removal efficiency of herbicide in ultrapure water (Figure 8b). As can be seen, the highest elimination efficiency was achieved in the presence of 19ZrO_2_/Fe_3_O_4_ nanopowder. Moreover, adsorption of herbicide on the surface of all three newly synthesized nanopowders was observed during 15 min of stirring in the dark. Namely, the adsorption rates were in the range of 30–40%.

Therefore, the influence of stirring/sonication on the efficiency of the heterogeneous photo-Fenton process was investigated (Appendix A). It is observed that in the case of fluroxypyr in the presence of 19ZrO_2_/Fe_3_O_4_/H_2_O_2_, the adsorption during 15 min of stirring is higher compared to sonication in the dark.

Also, the effect of the initial concentration of H_2_O_2_ (3.0 and 45 mmol/dm^3^) on the photocatalytic degradation of fluroxypyr was examined (Appendix A). As can be seen, the increased initial concentration of H_2_O_2_ from 3.0 to 45 mmol/dm^3^ was accompanied by increasing elimination efficiency of herbicide.

The efficiency of the photocatalytic degradation was investigated using TOC measurements, and it was found that for 240 min of irradiation, practically no fluroxypyr mineralization occurred.

#### 2.4.2. Photodegradation of Pharmaceutically Active Compound

Amitriptyline. Besides the pesticides, the photodegradation of amitriptyline from the group API was also studied. Amitriptyline is a tricyclic antidepressant from the class of dibenzocycloheptene and is most commonly used in the treatments of depression, migraines, chronic pain, fibromyalgia, neurological pain, etc. [42]. Amitriptyline is one of the most favored antidepressants, and it operates by inhibiting the uptake of serotonin and noradrenaline in the central nervous system [43]. In addition, it slightly inhibits the activity of dopamine, too [44]. The intensive consumption and the high persistence of antidepressants in the environment have a physiological impact on aqueous organisms, which is more than enough reason to monitor the occurrence of these compounds in nature [45]. When examining the efficiency of ZrO_2_/Fe_3_O_4_ nanopowders in the photocatalytic degradation of amitriptyline using solar irradiation, the efficiency of Fe_3_O_4_ was firstly investigated, with the pH adjusted to 2.8 and H_2_O_2_ added to the system at a concentration of 45 mmol/dm^3^. Based on the obtained results (Figure 9a), it can be concluded that the removal of amitriptyline significantly increased in the presence of Fe_3_O_4_ compared to the results of direct photolysis, and after 240 min of irradiation, 92% of amitriptyline was removed from the aqueous system. Certainly, the significant adsorption of amitriptyline on Fe_3_O_4_ should be taken into account, which amounts to 54% after 15 min of sonication in the dark. Also, the mineralization of amitriptyline, and possibly formed intermediates during the photocatalytic degradation of amitriptyline in the presence of pure Fe_3_O_4_, was monitored, and it was found that, after 240 min of irradiation, 8.3% of amitriptyline was mineralized.

Then, since in the case of thiacloprid 12ZrO_2_/Fe_3_O_4_ nanopowder proved to be the most efficient in photocatalytic degradation, its efficiency was tested at pH without adjustment (pH 5.0), as well as when pH was adjusted to 2.8 and added H_2_O_2_ (45 mmol/dm^3^). As can be seen in Appendix A, at pH 5.0, no amitriptyline degradation occurs, even though the efficiency of amitriptyline removal is lower compared to the direct photolysis process, while at pH 2.8 and with the addition of H_2_O_2_, significant amitriptyline removal was achieved, which after 240 min of irradiation was 78%.

In order to examine whether the influence of the mass percentage of ZrO_2_ in ZrO_2_/Fe_3_O_4_ nanomaterial on the efficiency of photocatalytic degradation also depends on the type of substrate, in addition to 12ZrO_2_/Fe_3_O_4_, the efficiency of 3.5ZrO_2_/Fe_3_O_4_ and 19ZrO_2_/Fe_3_O_4_ in the photocatalytic degradation of amitriptyline using solar radiation was examined (Figure 9b), at pH 2.8 and in the presence of H_2_O_2_ (45 mmol/dm^3^). Based on the obtained results, it can be noted that with an increase in the mass percentage of ZrO_2_, the adsorption of amitriptyline increases, as well as the percentage of its removal, and 19ZrO_2_/Fe_3_O_4_ proved to be the most effective nanomaterial. Based on our previous results [46], in which the photocatalytic degradation of amitriptyline was investigated in the presence of P25, a similar removal efficiency was found for the same irradiation time (60 min).

Given that 19ZrO_2_/Fe_3_O_4_ proved to be the most effective in amitriptyline degradation using solar irradiation, the influence of sample preparation (sonication or stirring) before irradiation, as well as the influence of H_2_O_2_ concentration (3.0 or 45 mmol/dm^3^) was examined (Appendix A). As is evident, when sonication was performed before irradiation in the presence of 45 mmol/dm^3^ H_2_O_2_, 29% of amitriptyline was adsorbed, while in the case of stirring on a magnetic stirrer, 38% of amitriptyline was adsorbed. In addition, the obtained results show that in the presence of a lower H_2_O_2_ concentration, the efficiency of amitriptyline removal decreased, and after 240 min of irradiation, 45% of amitriptyline was removed, while using 45 mmol/dm^3^ H_2_O_2_, for the same irradiation time, 95% of amitriptyline was removed from the reaction mixture. The degree of amitriptyline mineralization in the presence of 3.0 mmol/dm^3^ H_2_O_2_ was 12%, while in the presence of 45 mmol/dm^3^ H_2_O_2_, 17.5% of amitriptyline was mineralized after 240 min of irradiation.

### 2.5. The Influence of Natural Water Matrix on the Efficiency of Photocatalytic Degradation of Selected Pesticide and API Using 19ZrO_2_/Fe_3_O_4_ Nanopowder

In the example of sulcotrione (pesticide) and amitriptyline (API), the influence of natural water matrix on the efficiency of their photocatalytic degradation using 19ZrO_2_/Fe_3_O_4_ nanopowder was investigated.

Sulcotrione. The influence of the water matrix on the efficiency of sulcotrione photodegradation was investigated using 19ZrO_2_/Fe_3_O_4_ nanopowder with 3.0 mmol/dm^3^ H_2_O_2_ at pH 2.8 (Figure 10a). The obtained results indicate that sulcotrione almost equivalently behaved in the water from the Danube River and in ultrapure water. Namely, there were 92% (Danube River) and 96% (ultrapure water) sulcotrione removed after 240 min of irradiation. The adsorption was also similar; there was 35% of sulcotrione adsorbed in the system with Danube River water prior to irradiation, while in the case of ultrapure water, 39% of sulcotrione was adsorbed on the catalyst’s surface. On the other side, drinking and underground waters also have an alike effect on the removal of the mentioned pesticide. There was 21% of sulcotrione adsorbed in drinking water, while in underground water, the adsorption was 28%. These findings propose that the synthesized 19ZrO_2_/Fe_3_O_4_ nanopowders have real practical applications under natural conditions.

Amitriptyline. The influence of the water quality on the efficiency of the photocatalytic degradation of amitriptyline is presented in Figure 10b. As may be seen from the obtained results, in this case, the efficiency of amitriptyline removal increases when the photocatalytic degradation process takes place in drinking, Danube and groundwater, which also indicates the applicability of the tested system in real conditions. The percentage of amitriptyline removal in drinking water after 240 min of irradiation was 61%, in Danube River water, 61%, while in groundwater, it was slightly lower, 51%. In addition, the adsorption of amitriptyline on the 19ZrO_2_/Fe_3_O_4_ photocatalyst was slightly higher in drinking, Danube and groundwater compared to the results obtained in ultrapure water.

## 3. Materials and Methods

### 3.1. Chemicals and Solutions

All chemicals were of technical purity and were used with no additional purification. Thiacloprid ((Z)-3-((6-chloro-3-pyridinylmethyl)-1,3-thiazolidin-2-ylidenecyanami- de), CAS No. 111988-4998, C_10_H_9_ClN_4_S, *M*_r_ = 252.8) was determined from the commercial formulation Calypso^®^ 480–SC, Bayer (concentrated suspension, 480 mg/cm^3^ of thiacloprid) which was bought in a local store. The commercial formulation of neonicotinoid was used without former purification. Herbicides sulcotrione (2-(2-chloro- 4(methylsulfonyl)benzoyl)-1,3-cyclohexanedione, CAS No. 99105-77-8, C_14_H_13_ClO_5_S, *M*_r_ = 328.8, PESTANAL^®^, analytical standard, purity 99.9%) and fluroxypyr (4-amino-3,5-dichloro-6-fluoro-2-pyridyloxyacetic acid, CAS No. 69377-81-7, C_7_H_5_Cl_2_FN_2_O_3_, *M*_r_ = 255.03, PESTANAL^®^, analytical standard, purity 98.9%) are Fluka products. Pharmaceutically active compound, amitriptyline hydrochloride (3-(10,11-dihydro-5H-dibenzo[a,d][7]annulen-5-ylidene)-N,N-dimethylpropan-1-amine hydrochloride, CAS No. 549-18-8, C_20_H_24_ClN, *M*_r_ = 313.9, purity ≥ 98%) is a Sigma−Aldrich product.

The following chemicals were applied for the synthesis of ZrO_2_/Fe_3_O_4_ nanopowders: iron(III) chloride (FeCl_3_·6H_2_O, Poch, Gliwice, Poland, p.a.), iron(II) sulfate (FeSO_4_·7H_2_O, Poch, Gliwice, Poland, p.a.), zirconium(IV) oxide (Alfa Aesar, Karlsruhe, Germany, purity 99.8%) and sodium hydroxide (NRK Engineering, Belgrade, Serbia, purity 99.8%).

The rest of the chemicals were as well used without prior purification, with all of them being p.a. purity: 35% HCl and 85% H_3_PO_4_, Lachema, Neratovice, Czech Republic; NaOH, ZorkaPharm, Šabac, Serbia; 60% HClO_4_, Merck, Darmstadt, Germany; 99.9% acetonitrile C_3_H_3_NO (ACN for HPLC) ≥ 99.9% Sigma−Aldrich and H_2_O_2_ (30%, Sigma−Aldrich).

The Danube River water samples (Novi Sad, Serbia) and underground water samples (Štrand, Novi Sad, Serbia) were collected and were filtered through filter paper (Whatman, diameter 125 nm, pore size 0.1 µm), whilst drinking water samples were taken from the local water supply system (Novi Sad, Serbia). The physicochemical properties of examined waters and ultrapure water are presented in Appendix A.

To prepare the stock solutions of a concentration of 0.38/0.05 mmol/dm^3^, the proper mass of the studied substance was measured on a microscale and dissolved in the appropriate volume of ultrapure water. When the influence of the natural water matrix on the photocatalytic degradation of selected organic pollutants was examined, the experiments were performed by dissolving the appropriate mass of API, i.e., pesticides, on a microscale in the appropriate volume of natural water.

### 3.2. Catalyst Synthesis

Four ZrO_2_/Fe_3_O_4_ nanopowders with a different mass ratio of ZrO_2_ to Fe_3_O_4_ were synthesized via the chemical co-precipitation method. For each synthesis, 50 cm^3^ of 0.14 mol/dm^3^ FeSO_4_·7H_2_O aqueous solution and 100 cm^3^ 0.14 mol/dm^3^ FeCl_3_·6H_2_O aqueous solution (molar ratio Fe^2+^ to Fe^3+^ = 1:2) were mixed in a double-walled vessel. The mixture was stirred and warmed to 40 °C. Once this temperature was reached, the appropriate amount of ZrO_2_ was added. After 30 min, NaOH was dropped into the mixture until the pH reached 6.8. A large amount of black precipitate was then generated. The suspension was stirred for 1 h, and the obtained precipitates were filtered under a vacuum and washed with ultrapure water until no chloride was found in the filtrate (tested with AgNO_3_). The particles were dried in a desiccator under a vacuum with silica gel. Finally, the prepared composite particles were calcined at 300 °C for 3 h in static air. By applying this procedure, four samples with ZrO_2_ weight percent content of 0.9, 3.5, 12, and 19%, *w/w* (denoted as 0.9ZrO_2_/Fe_3_O_4_, 3.5ZrO_2_/Fe_3_O_4_, 12ZrO_2_/Fe_3_O_4_, and 19ZrO_2_/Fe_3_O_4_) were prepared. Photocatalysts with higher contents than 19 wt.% ZrO_2_ were not prepared because the excess loading may cover active sites on the Fe_3_O_4_ surface, thereby reducing photodegradation efficiency [47]. The same procedure, without the addition of ZrO_2_, was used for the synthesis of Fe_3_O_4_. It should be noted that the Fe_3_O_4_ label does not mean that the only formed Fe species was magnetite (especially taking into account the calcination step during the sample preparation). Based on XRD findings, the Fe_3_O_4_ labeled sample contains two Fe-oxide phases, hematite (α-Fe_2_O_3_) and magnetite (Fe_3_O_4_), in a ratio of about 0.3 to ~0.8.

### 3.3. Characterization Methods

Morphology of the synthesized nanopowders has been studied on SEM type JEOL JSM-6460LV with the operating voltage of 20 keV. The composition/quality has also been analyzed using the same SEM instrument equipped with an EDS INCAx-sight detectorand “INAx-stream” pulse processor (Oxford Instruments).

XRD was performed on a MiniFlex600 diffractometer (Rigaku, Japan) with Cu-K_α_ radiation (λ = 1.5406 Å) at a tube voltage of 40 kV and a tube current of 15 mA with a step size of 0.02° and a counting time of 1 °/min in the 2*θ* angular range from 10° to 80°.

Raman scattering experiments were performed using a Renishaw confocal Raman microscope Invia^TM^ (Gloucestershire, UK), where samples were excited with an argon-ion laser of 514 nm at ambient temperature. The laser beam was focused by a 50× objective. The Raman signal was collected by a CCD camera in the frequency range of 100–800 cm^−1^ with a spectral resolution of 2 cm^−1^. An accumulation time of 3 s was used for each spectrum.

The UV-vis DRS were obtained using an Ocean Optics QE65000 (Dunedin, FL, USA) high-sensitivity optical spectrometer in the wavelength range of 300–900 nm. The optical bandgaps were estimated from the Kubelka-Munk function using the SpectraSuite Ocean Optics operating software. All measurements were performed at room temperature.

### 3.4. Sample Preparation and Irradiation

Photocatalytic degradation was performed in a photochemical cell (total volume of approximately 40 cm^3^, liquid layer thickness 35 mm) made of Pyrexglass with double walls. The experiments were conducted using 20 cm^3^ of 0.38/0.05 mmol/dm^3^ solution of API/pesticide, with a mass concentration of 0.42/1.0 mg/cm^3^ for all nanopowders, except for the case when the optimal catalyst loading was examined. In the experiments with Fe_3_O_4_ and ZrO_2_/Fe_3_O_4_ (0.9, 3.5, 12, and 19%) nanopowders, the pHvalue was set with H_2_SO_4_ (0.5 mol/dm^3^) to 2.8, and H_2_O_2_ (3.0 or 45 mmol/dm^3^) was added, except for the case when the optimal H_2_O_2_ concentration was examined. The reaction mixture was sonicated (50 Hz) or stirred for 15 min in the dark before irradiation to achieve adsorption-desorption equilibrium. The photochemical cell was afterward placed onto a magnetic stirrer and thermostated at 25.0 °C and stirred in a stream of oxygen (3.0 cm^3^/min). During the irradiation period, the suspension was continuously stirred with a magnetic stirrer, and the O_2_ flow (3.0 cm^3^/min) was also continued, achieving its constant concentration during the irradiation. A halogen lamp (Philips, 50 W) was used as the source of solar irradiation Energy fluxes of UVA and Vis radiation was measured using a Delta Ohm (Padua, Italy) radiometer with sensors: LP 471 UVA (spectral area 315–400 nm) for the UVA region and LP 471 RAD (spectral area 400–1050 nm) for visible region. Photon flux for the halogen lamp was 63.85 mW/cm^2^ for visible irradiation and 0.22 mW/cm^2^ for the UVA region.

### 3.5. Analytical Procedures

To monitor the removal kinetics of pesticides and API from water, the aliquots of 0.5 cm^3^ of suspension were taken prior to the irradiation and at selected time intervals throughout the irradiation (volume variation ca. 10%). All obtained samples were then filtered through Millipore membrane filters (Millex-GV, 0.22 μm).

The degradation process of target compounds was monitored with liquid chromatograph Shimadzu UFLC with UV-vis DAD and column Eclipse XDB−C18 (150 mm × 4.6 mm, particle size 5 μm, 25 °C). Experimental conditions for the chromatographic analysis of the studied compounds are given in Table 3.

An ion chromatograph was used to determine the anion content in natural waters. A volume of 10 μL of the sample was injected and analyzed on an ion chromatograph Dionex ICS 3000 Reagent-Free IC with a conductometric detector (ASRS ULTRA II, 4 mm) and a column IonPac AS15 Analytical (250 mm × 4 mm, bead diameter 9.0 μm), together with a pre-column IonPac AG15 Guard (50 mm × 4 mm, bead diameter 9.0 μm). The elution type was gradient using KOH (38 mmol/dm^3^) as a mobile phase with a flow rate of 1.2 cm^3^/min at the working temperature of 30 °C. The parameters of the conductometric detection with the mobile phase conductivity suppression: power 113 mA, background conductivity 0.5−2 μS, noise < 5 nS/min, and pressure ~2500 psi. Cations were determined using IonPac CS12A analytical column (250 mm × 4 mm i.d., bead diameter 7.5 μm) and a conductometric detector. The mobile phase was a 40 mmol/dm^3^ methane-sulfonic acid solution with a flow rate of 1 cm^3^/min.

For the determination of TOC, aliquots of 10 cm^3^ were taken prior to and 240 min after the irradiation. The samples were afterward diluted with ultrapure water in 25 cm^3^ volumetric flasks, filtered through membrane filters and analyzed with ElementarLiqui TOC II instrument according to the standard method US 120 EPA Method 9060A. Directly before TOC determination, 0.05 cm^3^ of concentrated 35% HCl was added to the blank and the samples.

The change in pHvalue during the degradation was monitored using a combined glass electrode (pH-Electrode SenTix 20, WTW) connected to the pHmeter (pH/Cond 340i, WTW).

## 4. Conclusions and Outlooks

In this research, the possible rapid removal of the selected pollutants was investigated using newly synthesized ZrO_2_/Fe_3_O_4_ nanoparticles under solar irradiation. ZrO_2_/Fe_3_O_4_ nanoparticles with different ZrO_2_ content (0.9, 3.5, 12, and 19%, *w/w*) were synthesized by simple chemical co-precipitation. The presence of magnetite (Fe_3_O_4_) and hematite (Fe_2_O_3_) phases was confirmed in all samples. The content of the magnetite phase is the highest, with the addition of 19% ZrO_2_. The ZrO_2_/Fe_3_O_4_ morphology shows clusters of irregularly shaped particles between 55 and 198 nm in size, while the crystallite size does not change significantly with ZrO_2_ modification and is about 14 nm and about 6 nm for the magnetite and hematite phases, respectively. Raman spectroscopy showed a further transformation of the magnetite to the hematite phase induced by the high-power laser used to record the Raman spectra. The high visible absorption capacity of ZrO_2_/Fe_3_O_4_ nanoparticles for photocatalysis is predicted by DRS measurements.

The efficiency of the above-mentioned catalysts was examined in the systems with thiacloprid under various experimental conditions. According to the findings, it can be concluded that the highest removal was achieved in the 12ZrO_2_/Fe_3_O_4_/H_2_O_2_ photo-Fenton system after 120 min of solar irradiation. Contrary to that, in the case of sulcotrione, fluroxypyr, and amitriptyline, the 19ZrO_2_/Fe_3_O_4_ nanoparticles in the presence of H_2_O_2_ were found to be the most efficient. The TOC measurements confirmed the process of photocatalytic degradation and showed an acceptable mineralization rate of the selected emerging pollutants. Lastly, the effect of the water matrix was also examined. According to our results, the ions present in different water samples are capable of enhancing the efficiency of photocatalytic degradation, which can be explained by the formation of additional, highly reactive radicals.

In this study, we adequately recommended a promising removal technique for four different emerging pollutants from an aqueous environment. However, further steps should be taken in order to achieve a more competent and sustainable water remediation technique. To begin with, the synthesis of the ZrO_2_/Fe_3_O_4_ nanoparticles should be more eco-inspired and sustainable. Namely, new techniques should be developed using fewer chemicals, and greener approaches. Furthermore, the degradation pathways of the mentioned pollutants should be examined in detail in order to understand the photodegradation process and to get information about the degradation intermediates. In this way, we could endorse more suitable methods to completely mineralize the present pollutants. Nevertheless, further toxicology investigations should be also carried out to gain more information about the harmful effects of the selected pollutants. Finally, effective and low-cost separation techniques could be developed, as well. By doing this, the applied catalysts could be recycled and used again in heterogeneous photocatalysis.

## Figures and Tables

**Figure 1 molecules-27-08060-f001:**
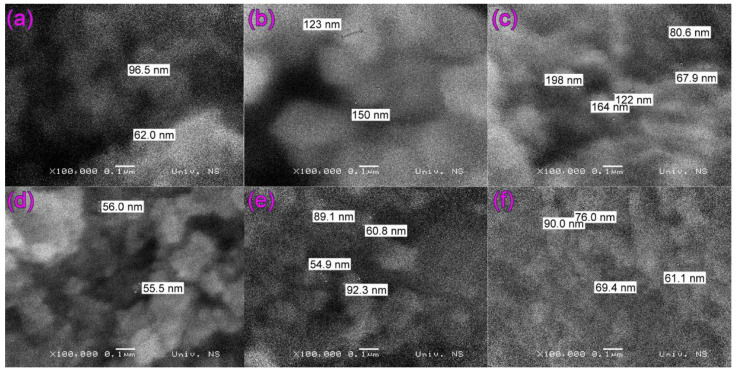
SEM images of particles: (**a**) Fe_3_O_4_ and (**b**) ZrO_2_, as well as synthesized nanopowders: (**c**) 0.9ZrO_2_/Fe_3_O_4_; (**d**) 3.5ZrO_2_/Fe_3_O_4_; (**e**) 12ZrO_2_/Fe_3_O_4_ and (**f**) 19ZrO_2_/Fe_3_O_4_. Line bar 0.1 μm.

**Figure 2 molecules-27-08060-f002:**
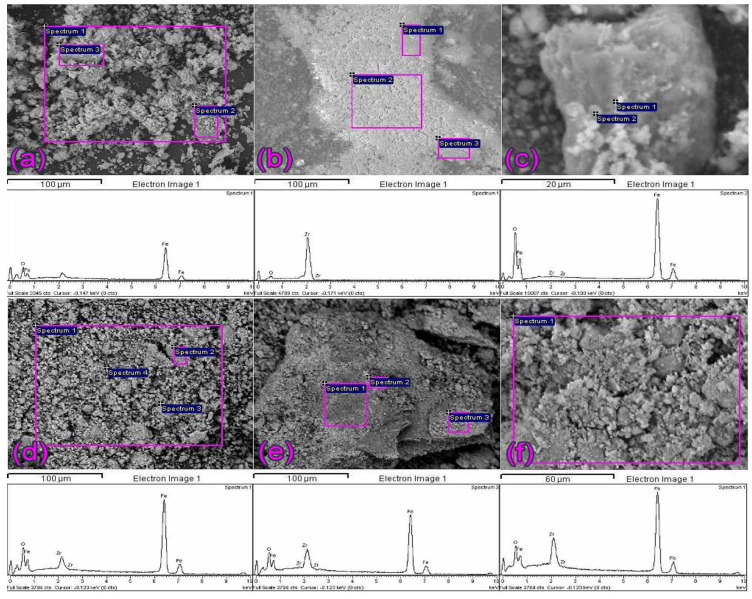
EDS spectra of particles: (**a**) Fe_3_O_4_ and (**b**) ZrO_2_, as well as synthesized nanopowders: (**c**) 0.9ZrO_2_/Fe_3_O_4_; (**d**) 3.5ZrO_2_/Fe_3_O_4_; (**e**) 12ZrO_2_/Fe_3_O_4_ and (**f**) 19ZrO_2_/Fe_3_O_4_, with corresponding area shown in SEM images.

**Figure 3 molecules-27-08060-f003:**
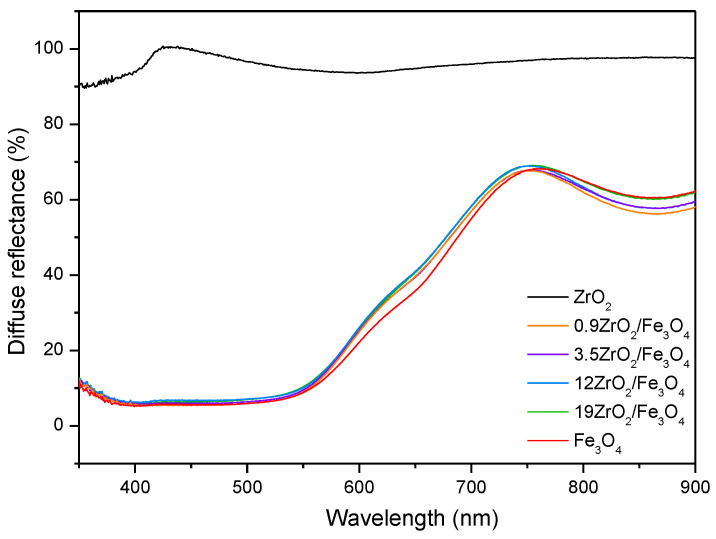
UV-vis DRS of particles Fe_3_O_4_ and ZrO_2_, as well as synthesized ZrO_2_/Fe_3_O_4_ nanopowders.

**Figure 4 molecules-27-08060-f004:**
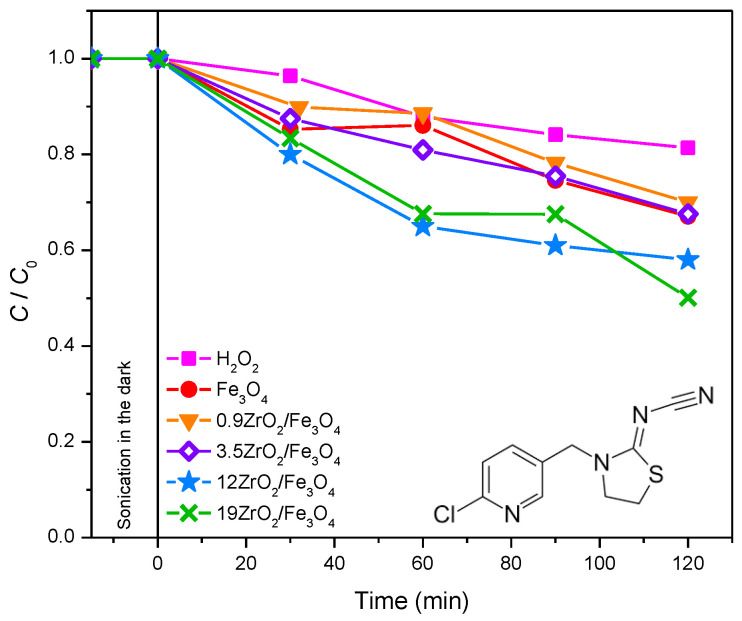
The influence of different photodegradation systems on the kinetics of thiacloprid (0.38 mmol/dm^3^) photodegradation in the presence of 45 mmol/dm^3^ H_2_O_2_ and 0.42 mg/cm^3^ catalyst at pH 2.8, using solar irradiation. Inset represents the structural formula of thiacloprid.

**Figure 5 molecules-27-08060-f005:**
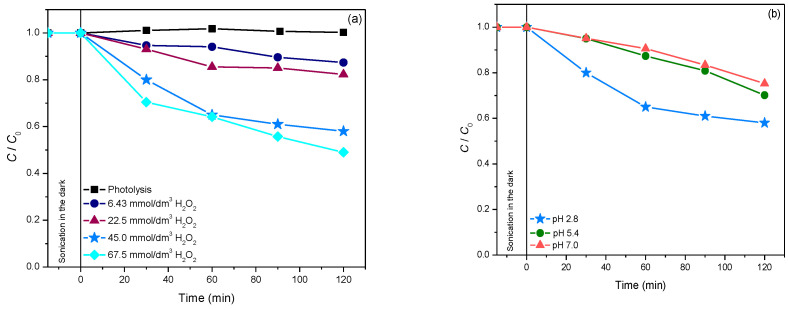
(**a**) The influence of initial H_2_O_2_ concentration on the kinetics of thiacloprid (0.38 mmol/dm^3^) photodegradation in the presence of 0.42 mg/cm^3^ 12ZrO_2_/Fe_3_O_4_ at pH 2.8 and using solar irradiation. (**b**) The influence of initial pH on the thiacloprid (0.38 mmol/dm^3^) photodegradation efficiency in the presence of 45 mmol/dm^3^ H_2_O_2_, 0.42 mg/cm^3^ 12ZrO_2_/Fe_3_O_4_ and using solar irradiation.

**Figure 6 molecules-27-08060-f006:**
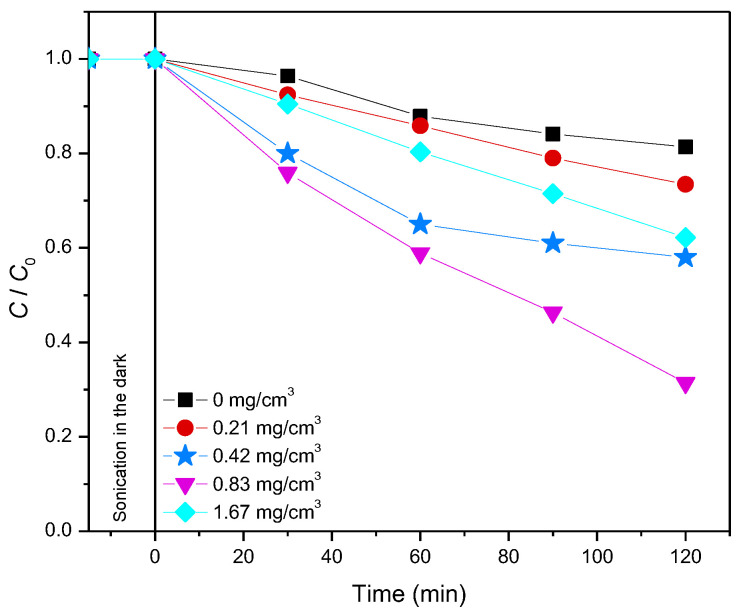
The influence of 12ZrO_2_/Fe_3_O_4_ concentration on the thiacloprid (0.38 mmol/dm^3^) photodegradation kinetics in the presence of 45 mmol/dm^3^ H_2_O_2_ at pH 2.8 and using solar irradiation.

**Figure 7 molecules-27-08060-f007:**
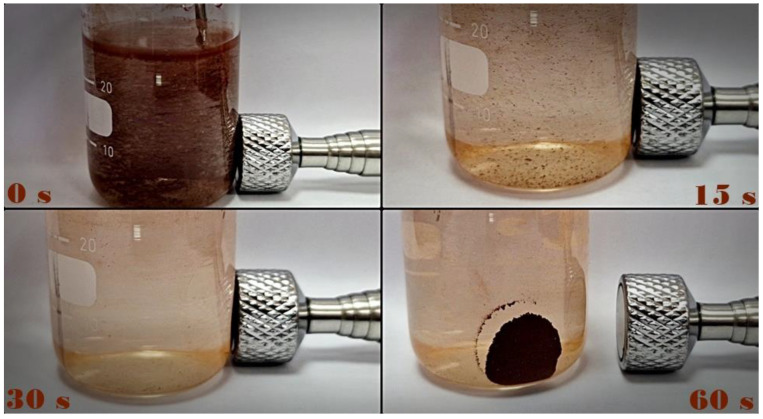
Suspension of 12ZrO_2_/Fe_3_O_4_ (1.67 mg/cm^3^) photocatalyst at selected time intervals throughout the application of an external magnetic field.

**Figure 8 molecules-27-08060-f008:**
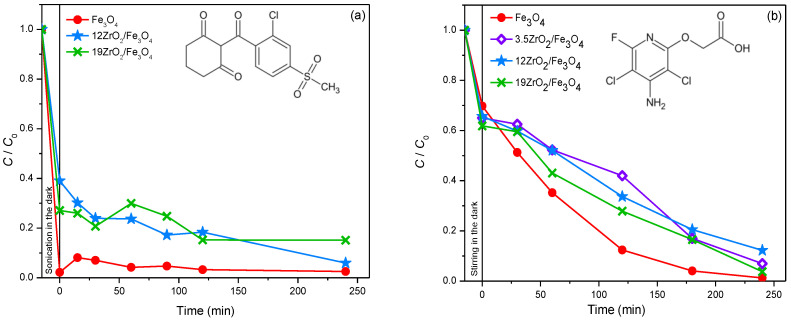
(**a**) Kinetics of removal of sulcotrione (0.05 mmol/dm^3^) at pH 2.8 in the presence of Fe_3_O_4_, 12ZrO_2_/Fe_3_O_4_ and 19ZrO_2_/Fe_3_O_4_ (1.0 mg/cm^3^) with 45 mmol/dm^3^ H_2_O_2_ and using solar irradiation. Inset represents the structural formula of sulcotrione. (**b**) Effect of stirring the suspension before irradiation on the adsorption and degradation kinetics of fluroxypyr (0.05 mmol/dm^3^) using Fe_3_O_4_ and different ZrO_2_/Fe_3_O_4_ nanopowders in the presence of H_2_O_2_ (3.0 mmol/dm^3^) at pH 2.8 and solar irradiation. Inset represents the structural formula of fluroxypyr.

**Figure 9 molecules-27-08060-f009:**
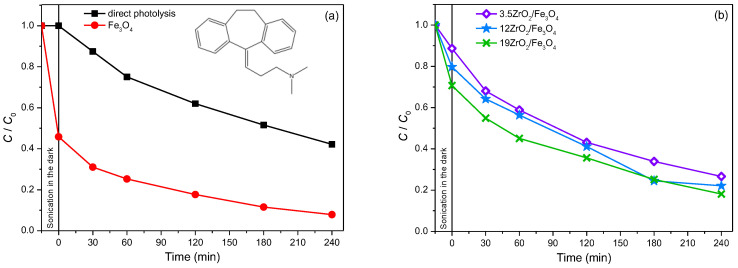
(**a**)Kinetics of photolytic and photocatalytic degradation of amitriptyline (0.05 mmol/dm^3^) in the presence of Fe_3_O_4_ (1.0 mg/cm^3^) as a photocatalyst, H_2_O_2_ (45 mmol/dm^3^), at pH 2.8 and using solar irradiation. Inset represents the structural formula of amitriptyline. (**b**) Kinetics of photocatalytic degradation of amitriptyline (0.05 mmol/dm^3^) in the presence of different ZrO_2_/Fe_3_O_4_ (1.0 mg/cm^3^) as photocatalysts, H_2_O_2_ (45 mmol/dm^3^), at pH 2.8 and using solar irradiation.

**Figure 10 molecules-27-08060-f010:**
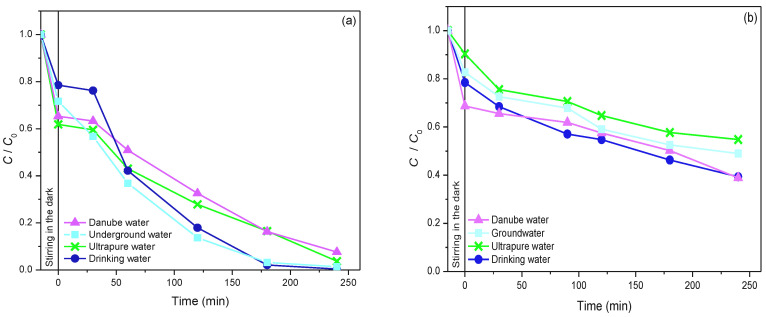
Influence of water matrix on the removal of sulcotrione (**a**) and amitriptyline (**b**) under the following experimental conditions: substrate (0.05 mmol/dm^3^) in the presence of 19ZrO_2_/Fe^3^O^4^ (1.0 mg/cm^3^) nanopowders with H_2_O_2_ (3.0 mmol/dm^3^) at pH 2.8, using solar irradiation.

**Table 1 molecules-27-08060-t001:** Mass composition of the materials characterized by the EDS method.

Sample	O (wt.%)	Fe (wt.%)	Zr (wt.%)	Total (wt.%)
Fe_3_O_4_	24.24	75.76	-	100.00
ZrO_2_	28.43	-	71.57	100.00
0.9ZrO_2_/Fe_3_O_4_	25.24	74.15	0.61	100.00
3.5ZrO_2_/Fe_3_O_4_	22.22	74.16	3.62	100.00
12ZrO_2_/Fe_3_O_4_	32.24	58.90	8.86	100.00
19ZrO_2_/Fe_3_O_4_	22.96	64.16	12.88	100.00

**Table 2 molecules-27-08060-t002:** Results of XRD measurements.

Sample	Crystallite Size (nm)		Hematite/Magnetite Ratio
	Hematite	Magnetite	
Fe_3_O_4_	4.6	13.0	0.45
0.9ZrO_2_/Fe_3_O_4_	7.5	14.7	0.59
3.5ZrO_2_/Fe_3_O_4_	5.1	14.3	0.77
12ZrO_2_/Fe_3_O_4_	6.0	12.3	0.75
19ZrO_2_/Fe_3_O_4_	8.2	16.2	0.31

**Table 3 molecules-27-08060-t003:** The experimental conditions for the isocratic chromatographic analysis of the studied compounds.

Compound	Mobile Phase Composition ACN:0.1% Water Solution H_3_PO_4_ (*v/v*) and pH of Mobile Phase	Injected Volume (μL)	Wavelength (nm) ^1^	Flow Rate (cm^3^/min)	Retention Time (min)
Thiacloprid	30:70, pH 2.25	20	242	0.8	8.5
Sulcotrione	50:50, pH 2.54	20	231	1.0	4.8
Fluroxypyr	50:50, pH 2.54	10	212	1.0	3.5
Amitriptyline	40:60, pH 2.50	10	206	0.8	5.9

^1^ The wavelength of the absorption maximum of the analyzed compound.

## Data Availability

The data presented in this study are available in Appendix A.

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
