# Peer review of "Rapid Removal of Organic Pollutants from Aqueous Systems under Solar Irradiation Using ZrO2/Fe3O4 Nanoparticles"

_molecules, 2022, doi:10.3390/molecules27228060_

Round 1
Reviewer 1 Report
ZrO2/Fe3O4 nanopowders with different mass ratio of ZrO2 and Fe3O4 were synthesized and were found to exhibit good catalytic performance in photocatalytic degradation of organic pollutants under solar irradiation. However, the innovation of this manuscript is weak since no discussion about the relationship between structure and ability was included. Therefore, major revisions are needed.
Here are some issues:
1) The author should give some explanation why ZrO2/Fe3O4 nanopowder is better as compared with Fe3O4.
2) 19ZrO2/Fe3O4 is the best among the four catalysts in most case. Thus, will the performance become better if further increasing the ZrO2 content?
3) The efficiency of the photocatalytic degradation of present catalysts should be compared with a reference one (e.g. P25).
Reviewer 2 Report
The manuscript entitled “Rapid removal of organic pollutants from aqueous systems under solar irradiation using ZrO2/Fe3O4 nanoparticles” is quite interesting and a fruitful approach for the removal of organic pollutants from aqueous system. The paper can be accepted after following main points.
1) The presentation of your work is quite poor, the introduction portion seems to be the discussion and quite long, on the other hand the result and discussion don’t have discussion portion. You need to resharpen the introductory and result & discussion portion well.
2) As presented Zirconium was identified in all samples (fig3c-d), but in the case of lower content, it was not identified. Explain it well as zirconium is the main element of your nanoparticles.
3) The introductory portion should contain the recent methods also that is used for such purpose, add the following references if you find it useful.
https://doi.org/10.3390/ijerph19169962 , https://doi.org/10.3390/w14101551
https://doi.org/10.3390/w13212959 , https://doi.org/10.1155/2020/8216435
4) The quality of EDS spectra figure is quite low, if possible, improve it

Round 2
Reviewer 1 Report
The paper can be published after the revision.